# Erythrocytes as Carriers: From Drug Delivery to Biosensors

**DOI:** 10.3390/pharmaceutics12030276

**Published:** 2020-03-18

**Authors:** Larisa Koleva, Elizaveta Bovt, Fazoil Ataullakhanov, Elena Sinauridze

**Affiliations:** 1Laboratory of Biophysics, Dmitriy Rogachev National Medical Research Center of Pediatric Hematology, Oncology, and Immunology, Ministry of Healthcare of Russian Federation, Samory Mashela str., 1, GSP-7, Moscow 117198, Russia; ie.bovt.rv@gmail.com (E.B.); ataullakhanov.fazly@gmail.com (F.A.); 2Laboratory of Physiology and Biophysics of the Cell, Center for Theoretical Problems of Physicochemical Pharmacology, Russian Academy of Sciences, Srednyaya Kalitnikovskaya, 30, Moscow 109029, Russia; 3Department of Physics, Lomonosov Moscow State University, Leninskie Gory, 1, build. 2, GSP-1, Moscow 119991, Russia

**Keywords:** drug delivery, erythrocyte, carrier erythrocyte, erythrocyte-bioreactor, targeted drug delivery, therapy, diagnostics

## Abstract

Drug delivery using natural biological carriers, especially erythrocytes, is a rapidly developing field. Such erythrocytes can act as carriers that prolong the drug’s action due to its gradual release from the carrier; as bioreactors with encapsulated enzymes performing the necessary reactions, while remaining inaccessible to the immune system and plasma proteases; or as a tool for targeted drug delivery to target organs, primarily to cells of the reticuloendothelial system, liver and spleen. To date, erythrocytes have been studied as carriers for a wide range of drugs, such as enzymes, antibiotics, anti-inflammatory, antiviral drugs, etc., and for diagnostic purposes (e.g., magnetic resonance imaging). The review focuses only on drugs loaded inside erythrocytes, defines the main lines of research for erythrocytes with bioactive substances, as well as the advantages and limitations of their application. Particular attention is paid to in vivo studies, opening-up the potential for the clinical use of drugs encapsulated into erythrocytes.

## 1. Erythrocytes as Drug Carriers

Drug delivery using natural biological carriers is a fast-developing field. Due to the unique biophysical properties, erythrocytes (red blood cells, RBCs) have great potential in this area. RBCs are the largest population of blood cells in mammals. Their main function is oxygen transfer to cells and body tissues [1]. Mature RBCs do not have a cell nucleus and most organelles, but they contain a large amount of a special protein, hemoglobin (Hb), which is able to bind to oxygen. The biconcave shape provides good flexibility and allows the erythrocyte to deform and pass through narrow capillaries. The lifetime of erythrocytes in the bloodstream is 100–120 days, after which they are removed by the spleen. Erythrocytes can be used as carriers in two different ways: by incorporating the drug into the cells or by binding it (using non-specific adsorption or a specific association, involving antibodies or various chemical cross-linking compounds) on the RBCs’ surface. Our review focuses on the first of these methods. The binding of drugs on the surface of RBCs has both advantages and disadvantages. A great contribution to the development of this direction was made by the team of Muzykantov et al. [2,3,4,5,6,7,8,9].

To incorporate the drug into the RBC, the cell must undergo some external influences so that pores can be reversibly formed in its membrane, through which the drug can penetrate. This unique property of RBCs allows to load them with biologically active substances of different molecular weights. For these reasons, erythrocytes are promising biocompatible cells for drug delivery.

The methods for incorporating various substances into red blood cells differ in the way that substances penetrate the cells. The cause of permeability may be the pores’ formation in the cell membrane due to a physical exposure (high voltage electric pulse [10,11] or ultrasound [12]). Drug molecules can also enter the RBCs by endocytosis in the presence of certain chemical compounds (for example, primaquine [13], vinblastine, chlorpromazine, hydrocortisone or tetracaine [14,15]), or using the cell-penetrating peptides bounded to the compound that should be encapsulated [16]. However, the most popular are different variants of osmotic methods.

In some cases, RBCs are first exposed to a hyperosmotic pulse of a low molecular weight substance that penetrates very well through the cell membrane (for example, dimethyl sulfoxide (DMSO) [17,18] or glucose [19,20]). After washing the cells, which decreases the external concentration of these compounds and creates a gradient of their concentration between both sides of the RBC membrane, the target drug is introduced into the external volume. Water with this drug begins to enter into the cells to decrease the osmotic pressure there. The process ends when the gradient of DMSO or glucose disappears. The pores close and part of the drug remains into RBCs. Other, the most popular of the osmotic methods are hypoosmotic. These methods are based on creating a hypotonic environment around RBCs, which causes swelling of the cells and opening pores in the cellular membrane, through which therapeutic compounds can penetrate RBCs. Then, a hypertonic solution is introduced into the cell suspension. The pores close, the cells restore their original size, trapping the drug molecules inside the cell. Osmotic methods are divided into several types. Simple reversible cell lysis in a hypotonic solution by dilutiing a cell suspension with a hypotonic medium causes the formation of erythrocyte ghosts [21,22]. The method of hypotonic pre-swelling is based on the initial controlled cells swelling in a hypotonic solution and their subsequent lysis by adding small portions of an aqueous solution of the drug for encapsulation [23,24,25]. Dialysis methods are based on a reduction of osmolality around the RBCs by a process of dialysis versus hypotonic solution in a dialysis bag [26,27] or in special dialyzers with increased area of contact of RBCs with a buffer solution in the case of flow dialysis [28,29,30]. As mentioned above, hypoosmotic methods are most preferable for incorporation of enzymes into RBCs in terms of efficacy and the properties of obtained cell carriers [31,32].

The history of carrier erythrocytes begins in 1973, when Ihler demonstrated in his article the possibility of incorporating enzymes such as β-glucosidase and β-galactosidase into these cells by reversible hypoosmotic lysis [21]. The analysis of the number of publications (relating only to medications inside the RBCs) shows that interest in this topic since 1973 has not declined, but, instead, has been constantly growing. The number of published articles on the subject of carrier erythrocytes increases every year, and currently, their total number is about 400 (Figure 1).

RBCs for drug delivery have several advantages compared to the existing methods and systems for drug delivery. The erythrocyte is an ideal candidate for such delivery and meets all the requirements for such systems, namely:-biocompatibility (human, both autologous and donor erythrocytes are used to treat patients);-biodegradability (old or damaged erythrocytes are naturally removed by the reticuloendothelial system);-long life in the bloodstream (the drug has an extended lifetime inside the cells because RBCs protect it from the immune system and plasma proteases and the cells survive in the body for a long time; thus, the pharmacokinetics and pharmacodynamics of the drug in RBCs can significantly increase the desired therapeutic effect);-decreasing side effects of drugs (due to preventing allergic reactions, and the decrease in the peak concentrations of free drug in the blood to safer levels);-ease of cell isolation in large quantities and the ability to scale production.

Carrier erythrocytes (CEs) can be used both in therapy and the diagnosis of some diseases, for example, as carriers of contrast agents for magnetic resonance imaging (MRI) or as biosensors that respond to changes in the concentration of metabolites or pH in the blood [33,34,35]. In therapy, depending on the drug that is loaded, the erythrocytes can be used as carriers with a gradual drug release, as bioreactors or a system for targeted drug delivery, primarily to the reticuloendothelial system (RES), liver and spleen [36]. In the first case, either a drug encapsulated into RBCs can slowly pass through the erythrocyte membrane into the bloodstream, or a membrane-nonpenetrating prodrug is loaded into RBCs, where it turns into a therapeutically effective compound that is able to exit the cell. This ensures prolonged drug circulation in the bloodstream with a decrease in the toxic effects on the body. In the second case, the enzyme encapsulated in erythrocytes works with substrates penetrating the cell membrane. Thus, the enzyme does not directly enter the bloodstream, which solves the problem of its immunogenicity, premature inactivation and increases its half-life.

In this review, we analyzed and organized all existing information on CEs with encapsulated bioactive substances, starting from 1973, that we found in the literature. A summary diagram of their possible use is presented in Figure 2. The most interesting and significant studies in this area are described below. Since there are separate articles in this Issue devoted to a detailed description of the enzymes loaded in RBCs, in our review this subject is considered very briefly (section Erythrocytes-bioreactors). For the most part, only the names of the enzymes that were incorporated into RBCs are listed to ensure the integrity of the review.

## 2. Erythrocytes-Bioreactors

CEs can operate as bioreactors when the enzyme is incorporated into RBCs. A loaded enzyme can remove the appropriate substrate from the bloodstream, provided that this substrate is able to penetrate into RBCs from the blood. Such erythrocytes-bioreactors (EBRs) open up new possibilities in the treatment of diseases associated with inborn deficiencies of enzymes (enzyme replacement therapy), in the treatment of malignant tumor diseases and for the removal of some toxic compounds from the bloodstream.

### 2.1. Enzyme Replacement Therapy

Many human diseases are associated with the absence or decrease in the activity of certain enzymes. The logical method for solving this problem is therapy based on the administration of the missing enzyme into the body. However, the injection of free enzyme into the blood, as a rule, is accompanied by the body’s immune response and rapid drug removal from the bloodstream. Incorporating an enzyme into RBCs may be a good solution in this situation. An increase in the blood half-life and a decrease in the body’s immunological reactions to the drug were shown for all enzymes encapsulated in RBCs.

Lysosomal storage diseases [37] (Gaucher disease [38,39], Slay syndrome [40], Fabry disease [41,42] or Pompe disease [43]) are caused by a deficiency of lysosomals enzymes such as of β-glucocerebrosidase (β-glucosidase) [21,26,38,39,44,45], β-glucuronidase [40], α-galactosidase [21] or α-glucosidase [21], respectively. Deficiency of lysosomal enzymes results in the gradual accumulation of their substrates in lysosomes, which ultimately leads to disruption of lysosomes, dysfunction and cell death. The β-glucocerebrosidase enzyme was the first that was incorporated into RBCs for use in enzyme replacement therapy. For an enzyme loaded in RBCs the four–five-fold increase in circulation time was observed [45,46]. Moreover, it was suggested that if RBCs loaded with β-glucocerebrosidase are modified by γ-globulin, then in the body, they must be captured by macrophages and, thus, delivered directly to the focus of the disease—RES cells [45]. Currently, a number of free lysosomal enzymes are used to treat lysosomal storage diseases, but they have high immunogenicity and high cost. Loading appropriate enzymes into RBCs can overcome these limitations and decrease the total cost of treatment.

Other enzymes that have been described for use in enzyme replacement therapy are phenylalanine hydroxylase (for phenylketonuria [47,48,49,50]), adenosine deaminase (for severe combined immunodeficiency with impaired humoral and cellular immune response [27,51,52,53,54,55]) and thymidine phosphorylase (for the treatment of mitochondrial neurogastrointestinal encephalomyopathy (MNGIE) [56,57]). For adenosine deaminase, successful long-term (9 years) use of the adenosine deaminase-loaded RBCs in the clinic has been described [58]. Thymidine phosphorylase and adenosine deaminase encapsulated into RBCs can be used as maintenance therapy before transplantation of allogeneic hematopoietic stem cells. These enzyme preparations are a less expensive alternative to the pegylated forms of the drugs used today.

### 2.2. Erythrocytes-Bioreactors for Low Molecular Metabolites Utilization

EBRs for removal of low molecular metabolites (ethanol, methanol, cyanide, glucose or ammonium) from bloodstream have been described. These EBRs were based on alcohol dehydrogenase [11,59,60], alcohol oxidase [61], acetaldehyde dehydrogenase [11,62] or alcohol- and acetaldehyde dehydrogenase together [63] in cases of ethanol, methanol and acetaldehyde removal.

Hexokinase and glucose oxidase (both separately and together) were used to remove of excess glucose. In the latter case, this allowed for the rate of glucose consumption in mice to be increased by almost 5.5 times and maintain its normal level for several weeks [64]. Rhodonase (a mitochondrial enzyme responsible for the transformation of cyanide into thiocyanate) was used in the presence of a sulfur donor (sodium thiosulfate or other) for cyanide detoxification [65,66,67,68,69,70]. In mice, it was shown in vivo that erythrocytes loaded with rhodanase in tandem with thiosulfate decreased the blood concentration of cyanide by 40% in 15 min [70].

Moreover, the use of EBRs loaded with asparaginase, methioninease (methionine-γ-lyase) and arginine deiminase for antitumor therapy has been described (see below).

#### Ammocytes

It would be interesting to go into more detail on the use of EBRs to remove excess ammonium from the bloodstream, since success in this direction has been demonstrated in the work of recent years. An immediate consequence of an ammonium excess in the blood (hyperammonemia) is encephalopathy with the possibility of a lethal outcome. Long-term low-degree hyperammonemia may be associated with neurodegenerative diseases, such as Alzheimer’s disease, Parkinson’s disease, etc. [71]. This condition can be caused by both hereditary deficiencies in the enzymes of the uric acid cycle, for example, arginase, and chronic or acute liver diseases. Maintaining low levels of ammonium in the blood is important for treating hyperammonemia and preventing or slowing the development of neurodegenerative diseases. Modern pharmaceutical approaches to reduce the level of ammonium in the blood, unfortunately, do not provide a satisfactory solution to this problem from the point of view of effectiveness and side effects. EBRs for ammonium removal (so-called ammocytes) have been developed by various scientific groups. For enapsulation into RBCs, glutamate dehydrogenase was used, which catalyzed the formation of l-glutamic acid from α-ketoglutarate and ammonium in the presence of NADPH [59,60,72], as well as glutamine synthetase, which catalyzed the formation of l-glutamine from l-glutamic acid and ammonium in the presence of ATP [73,74]. Each of these enzymes was encapsulated into RBCs using reversible hypoosmotic dialysis. However, in vivo experiments showed that such bioreactors effectively removed ammonium from the circulation in mice only in the first 0.5–1 h [72,74]. After this time, the concentration of ammonium in the blood decreased at about the same rate in both experimental and control animals that received dialyzed erythrocytes, but without encapsulated enzymes. Thus, after 0.5–1 h, the loaded enzymes ceased to contribute to the process of ammonium consumption. Using mathematical models of EBRs created in [75], it was shown that the reason for this behavior is the depletion of the substrates inside the cell (l-glutamic acid or α-ketoglutarate), which are consumed during the utilization of ammonium by these enzymes, but are unable to enter the cell from the bloodstream. The authors of [75] proposed a new promising system to create ammonium-removing EBRs, based on the RBCs entrapment of a tandem from two enzymes—glutamate dehydrogenase and alanine aminotransferase. As a result, a new metabolic pathway was created in the erythrocytes, in which α-ketoglutarate and l-glutamic acid were produced and consumed in a cyclic process. Thus, the problem of depletion of these substrates inside the cell was solved, and the system became independent of their transport. The in vivo consumption rate of ammonium in mice for such bioreactors was 2 mmol/(h×l_EBRs_). Moreover, they continued to work even 2 h after the administration, which distinguished them from the bioreactors described previously in the literature [72,74]. The authors of [75] calculated that under physiological conditions transfusion of 200 mL of such EBRs to a patient will lead to a decrease in the plasma ammonium concentration by 6 mM/day, which is 10 times higher than similar values (600 μM/day) for the best drugs to reduce ammonium concentration currently available.

### 2.3. Enzymes Used in Antitumor Therapy

l-asparaginase, methioninase and arginine deiminase decrease the blood level of amino acids (asparagine, methionine or arginine, respectively) necessary for cells for biosynthesis during division. This depletion acts more efficiently towards some lines of tumor cells, which cannot synthesize asparagine or arginine on their own (since they do not contain asparagine synthetase [76] or do not express the enzymes necessary for the intermediate stages of arginine synthesis [77,78,79]). Moreover, tumor cells divide much faster than normal ones [80]. In all cases, the encapsulation of enzymes into RBCs may be a suitable alternative to the pegylated forms of these enzymes that are used currently in therapy to increase the half-life and decrease the immunogenicity of these proteins.

ERYTECH Pharma has patented and conducted clinical trial of asparaginase in RBCs (GRASPA) for the treatment of acute lymphoblastic leukemia (ALL) and acute myeloid leukemia [81,82,83], and is currently conducting clinical trials of asparaginase-loaded RBCs (Eryaspase) for the treatment of metastatic pancreatic cancer (trial TRYbeCA-1) [84,85,86] and of triple-negative breast cancer (trial TRYbeCA-2). Eryaspase has proven to be especially effective in pancreatic cancer treatment in combination with chemotherapy [85,86]. Phase 2 clinical trials demonstrated that chemotherapy treatment with Eryaspase reduces the risk of mortality by 40% compared with chemotherapy treatment alone. This is the first case in clinical practice where l-asparaginase therapy has proven effective in treating a solid tumor.

The use of methioninase encapsulated in erythrocytes (erymethioninase) has been demonstrated in vivo in mice with glioblastoma [87] or with breast carcinoma [88]. In both cases, there was a significant decrease in tumor volume, prolonged depletion of methionine and good tolerance of loaded methioninase. In addition, the possibility and effectiveness of a combination of erymethioninase therapy with cancer cell immunotherapy to block the immune control points of PD-1 (anti-PD-1 therapy) was first demonstrated by Sénécha et al. [88]. Significant inhibition of tumor growth was noted and the survival time was increased for erymethioninase therapy in tandem with immunotherapy, compared with each therapy separately.

In 2015, ERYTECH Pharma patented the use of RBCs containing arginine deiminase (ERY-ADI) for the treatment of, in particular, hepatocarcinoma and malignant melanoma [89]. It was shown in mice that the time of arginine depletion (5 days) with ERY-ADI treatment was increased compared to the same time for the free form of ADI (24 h) [90].

### 2.4. Inositol Hexaphosphate in Erythrocytes

Sickle cell anemia (sickle cell disease, SCD) is a hereditary disease in which anemia develops and RBCs are sickle-shaped. The cause of anemia is the presence in the cells of an altered form of Hb. This is HbS that has an increased tendency to polymerization in capillaries under conditions of partial deoxygenation. Such HbS can polymerize and precipitate inside cells under deoxygenation conditions, forming strands. As a result, the cells acquire a sickle shape and are destroyed [1]. This process may be partially reversible if the cell suspension is re-oxygenated. To improve the condition of patients with SDC, various research groups have proposed modifying donor RBCs for transfusion by incorporating inositol hexaphosphate (IHP) [17,91,92,93,94]. Such CEs do not contain a loaded enzyme, but contain an allosteric effector of the main erythrocyte protein—Hb; therefore, they can also be conditionally called bioreactors. This effector binds to Hb 1000 times stronger than 2,3-dysphosphoglycerate and reduces the affinity of oxygen to Hb, which leads to a two- to three-fold increase in the ability of such erythrocytes to give back bound oxygen.

Bourgeaux et al. proved in in vitro experiments that the addition of erythrocytes loaded with IHP (IHP-RBC) to the blood of patients with SCD was seven times more effective at decreasing the number of sickle cells after deoxygenation and subsequent reoxygenation of the cells compared with the addition of unmodified normal RBCs to this blood [92]. In vivo, in a transgenic mouse model (BERK) that mimics human SCD, four repeated injections of IHP-RBCs were shown to improve overall survival, prevent severe anemia and significantly reduce the risk of vascular occlusion in mice [95]. Thus, in vitro and in vivo studies indicate the therapeutic potential of IHP-RBCs in sickle cell anemia.

## 3. Carrier Erythrocytes with a Gradual Release of the Pharmacological Agent

An erythrocyte loaded with a pharmacological substance is not necessarily a bioreactor. In some cases, such an RBC can act as a system with a gradual release of the drug into the bloodstream. This approach can be useful when it is necessary to maintain a constant therapeutic drug concentration in the blood for a long time and decrease its peak concentration immediately after drug administration. As a rule, this principle of CEs’ action works when low-molecular-weight substances are loaded in the erythrocyte.

### 3.1. Cytotoxic Drugs in Erythrocytes

#### 3.1.1. Anthracycline Antibiotics

More than 50 years ago, it was shown that anthracycline antibiotics (daunomycin, doxorubicin) have antitumor activity both for solid tumors and for acute lymphoblastic and myeloid leukemia [96]. Currently, anthracycline antibiotics are used in the complex treatment of many types of cancer. The mechanism of anthracycline antibiotics’ action is the inhibition of topoisomerase II due to the embedding of anthracycline between adjacent pairs of DNA bases, which causes the production of free hydroxyl radicals that adversely affect both the tumor and healthy tissues [97]. Myocardial tissue is particularly affected. The cardiotoxicity of the anthracyclines, which has a cumulative dose-dependent nature, has been shown in many works [98,99,100].

Doxorubicin (adriamycin) is a 14-hydroxidaunorubicin that was isolated from a mutant *Streptomyces peucetius* (var. *Caesius*), obtained from a daunorubicin-producing organism, *S. Peucetius*. Its preclinical therapeutic index was better than that of daunorubicin [101], but the number of side effects did not decrease [102]. Cardiotoxicity caused by anthracycline can be decreased or prevented by a regimen of administration that gives low peak plasma concentrations of the drug; therefore, the search for special carriers of anthracyclines is an urgent task.

Since the 1980s, various groups of authors from the USA, Russia and Japan loaded daunomycin and doxorubicin into RBCs by various methods. Tonetti et al. have shown that encapsulation of daunorubicin into RBCs can be achieved by simple diffusion of the drug through the erythrocyte membrane [103]. In vitro, it has been shown that treatment with glutaraldehyde of RBCs loaded with daunorubicin significantly slows the release of the drug from the cells compared with untreated RBCs. After cells’ incubation for 24 h at 37 °C, the amount of daunorubicin in the cells was 66% and 10% of the initial encapsulated concentration for the treated and untreated cells, respectively. Later, in dogs, in vivo, a significant decrease in the peak plasma concentration of doxorubicin for doxorubicin loaded into RBCs (18.2 ng/mL) compared to its free form (330 ng/mL) was shown, as well as the possibility of targeted drug delivery to the liver [104,105,106].

Pilot studies of RBCs loaded with anthracycline antibiotics in patients with leukemia and lymphomas were carried out by a group of Russian authors (Ataullakhanov et al.). They demonstrated the advantages of the administration of daunorubicin- and doxorubicin-loaded RBCs compared with the administration of the free medicines [107,108,109,110]. Both drugs in the RBCs were clinically effective and were better tolerated by patients. A decreased number of adverse reactions, a significant decrease in cardiotoxicity (with the absence of a cumulative effect), as well as an at least two-fold decrease in the peak concentration of drugs in plasma was observed. The half-life of drugs in the bloodstream was increased. The pharmacokinetics of doxorubicin demonstrates two phases—fast and slow. For the free form of doxorubicin, the concentration of the drug rapidly decreased within 10–30 min after administration, and after 12–24 h the concentration decreased to zero. A similar picture was observed for doxorubicin in erythrocytes in the fast phase, but after the plasma doxorubicin concentration decreased to 0.1 μg/mL, its level remained almost constant up to 3 days [109]. In a recent paper [111], the cardiotoxicity of doxorubicin in carrier erythrocytes obtained by electroporation was studied in healthy mice. All parameters related to cardiac function in mice treated with doxorubicin in erythrocytes were similar to those in the control group of healthy animals that were not injected with the drug, while the same parameters for mice treated with a free form of doxorubicin were significantly worse than in those of the control.

In 2006, a new synthetic anthracycline antibiotic, mitoxantrone, was encapsulated into RBCs. Its effectiveness is higher than that of doxorubicin; however, the use of mitoxantrone is limited by the high cardio- and nephrotoxicity of the drug. In that work, the optimal conditions for the entrapment of the drug in RBCs were selected and the possibility of incorporating sufficiently high doses of mitoxantrone without observing a damaging effect of the drug on the cells was shown. The encapsulation of this antibiotic in RBCs opens-up prospects for its use in clinics [112,113].

#### 3.1.2. Terpene Indole Alkaloids

Vincristine and vinblastine are alkaloids isolated from the plant *Cantharanthus roseus G. Don (Vinca rosea Linn.*), which show an antitumor and hypoglycemic activity. Their antitumor activity was discovered in the 1960s [114,115,116,117]. The mechanism of the antitumor effect of vincristine and vinblastine is associated with inhibition of microtubule polymerization due to the binding of alkaloids to tubulin. This interferes with cell division (both tumor and normal). Currently, these alkaloids remain the most-used class of anticancer drugs and are important components of standard chemotherapy regimens [118,119]. However, both drugs have a number of serious side effects that limit the possible administered dose [119]. Vinblastine toxicity includes bone marrow suppression (which limits the dose), gastrointestinal toxicity and strong extravasation (leakage of a drug from a vein into surrounding tissues) with the appearance of blisters, deep ulcers and tissue necroses. The main side effects of vincristine are peripheral neuropathy, hyponatremia, leukopenia, thrombocytopenia and hair loss. In addition, both drugs are carcinogenic and mutagenic. Drug resistance to both drugs is also common, which interferes with therapy.

Halahakoon et al. suggested that encapsulating vincristine and vinblastine into RBCs could partially solve the problem of side effects by reducing the peak concentration of drugs in the bloodstream. The authors loaded the drugs into RBCs by hypoosmotic stepwise lysis (pre-swelling) [120]. In vitro experiments showed that vincristine and vinblastine are released from CEs during incubation at 37 °C (in autologous plasma or isotonic buffer) at a rate of 100 μg/h, and about 50% of the drugs are released from CEs after 6 h of incubation [121]. Unfortunately, in vivo experiments are not yet available; therefore, it is difficult to judge the pharmacokinetics and the possible effectiveness of the erythrocyte carriers of these drugs. Currently, the topic of the encapsulation of terpene indole alkaloids in RBCs remains open and little studied.

### 3.2. Glucocorticoids

Glucocorticoids are steroid hormones synthesized by the adrenal cortex. Since the 1940s, natural and synthetic glucocorticoids have been widely used in various fields of medicine. They have anti-inflammatory, desensitizing, anti-allergic immunosuppressive, anti-shock and anti-toxic effects. However, prolonged use of steroids leads to serious side effects. The most commonly used systemic glucocorticoids are hydrocortisone, prednisolone, methylprednisolone and dexamethasone. These glucocorticoids have good oral bioavailability and are excreted mainly due to liver metabolism and renal excretion [122]. Frequent and high doses of glucocorticoids cause hormonal dependence and serious side effects, such as immune suppression and diabetes [123,124,125,126,127]. Since glucocorticoids are rapidly eliminated from the body (within 3–4 h), the drug should be taken several times a day to maintain the therapeutic dose [122]. A good solution to the problem of the drug’s rapid elimination is to create a carrier that gradually releases the drug into the bloodstream.

EryDel (Italy) created erythrocytes-carriers of glucocorticoids, in particular, with dexamethasone-21-phosphate. In 1997, D’Ascenzo et al. created CEs with the gradual release of dexamethasone and prednisolone by incorporating their prodrugs (dexamethasone-21-phosphate and prednisolone-21-phosphate) into RBCs by hypotonic dialysis. The encapsulation yield was 30% and 28% for dexamethasone-21-phosphate and prednisolone-21-phosphate, respectively. In the cell, dexamethasone-21-phosphate and prednisolone-21-phosphate are dephosphorylated by phosphorylases present in the erythrocyte and are gradually released from the cell by diffusion [128]. Recently, EryDel has conducted numerous clinical trials of dexamethasone-21-phosphate (Dex 21-P) in autologous RBCs (Ery-Dex), which have proven the advantages of using dexamethasone in RBCs over the free form of the drug. To investigate the safety and tolerability of the drug, patients with cystic fibrosis in the first part of the study [129] received increasing doses of ERY-Dex. To evaluate the effectiveness of long-term continuous release of low doses of dexamethasone in the bloodstream, in the second part of the study, nine patients received Ery-Dex at 4-week intervals for 15 months [129]. After repeated injections, a slow and prolonged delivery of dexamethasone into the bloodstream of up to 28 days was observed. It was shown that with prolonged use of Ery-Dex, very low doses of glucocorticoids provide a significant improvement in one of the indicators for diseases of the lungs or bronchi—FEV1. This is the maximum volume of air exhaled in 1 s after the deepest inhalation. There was also a significant reduction in infectious relapses caused by *Pseudomonas aeruginosa*, and the absence of side effects.

Positive results of the use of Dex 21-P loaded in autologous RBCs have also been shown for steroid-dependent patients of different age groups with Crohn’s disease. The absence of both side effects and the need to take steroid drugs, in addition to patients going into clinical remission were shown [130,131,132]. Similar efficacy (achieving remission and the absence of side effects) was observed in patients with ulcerative colitis [133]. Six-month treatment of patients with steroid-dependent ulcerative colitis using a low dose of Dex 21-P in autologous RBCs allowed for the abolition of oral steroids in most patients without steroid-related side effects, while maintaining clinical remission [133]. EryDel is currently conducting Phase III clinical trials of Ery-Dex (trials.gov NCT02770807) for patients with a rare hereditary disease—ataxia telangiectasia (AT) or Louis–Bar syndrome, for which there is currently no effective treatment. AT is a rare hereditary neurodegenerative disease caused by mutations in the ATM gene (Ataxia Telangiectasia, Mutated), which encodes a protein of the same name, whose role is to coordinate cell signaling pathways in response to double-stranded DNA breaks, oxidative stress and other genotoxic stress [134]. The disease is primarily characterized by cerebellar degeneration, telangiectasia, immunodeficiency, susceptibility to cancer and radiation sensitivity. The results of a Phase II clinical trial for 22 patients with AT who received Ery-Dex for 6 months were published in [135]. They showed good drug tolerance and the possibility of slowing the natural progression of the disease.

In parallel with the development of a new dosage form of dexamethasone, EryDel invented and patented an automatic device, the Red Cell Loader (RCL), which allows small molecules and proteins to be incorporated into RBCs by gradual hypoosmotic swelling of RBCs [136].

### 3.3. Insulin in Erythrocytes

Few studies have been devoted to the encapsulation of insulin into RBCs, probably because it has been shown that insulin inside RBCs loses activity [137,138]. It was shown that due to inactivation, the percentage of insulin incorporation into RBCs is only 4.8%–6% of the initial amount [138,139,140]. The amount of insulin in the cells can be stabilized, if inhibitors of its degradation are loaded in the RBCs together with insulin. Despite this, there are very little data available. In addition, the work of such cells in vivo has not been investigated [139].

In [140], the half-elimination time of insulin from rabbit bloodstream was compared in the case of intravenous and subcutaneous administration of a free form of porcine insulin or intravenous administration of this insulin loaded in rabbit RBCs. The efficiency of glucose removal using these forms of insulin from the bloodstream of normal and diabetic rabbits was also studied. The plasma half-life for the encapsulated insulin was almost two-times longer than for the free form (12 and 7 min, respectively). The difference between the initial and minimum glucose concentrations achieved during the administration of different forms of insulin was 95 ± 12, 53 ± 11 and 98 ± 19 mg/dL, for normal rabbits, and 304 ± 26, 488 ± 68 and 532 ± 57 mg/dL for rabbits with diabetes for the free form, administered intravenously and subcutaneously, and for insulin in RBCs, respectively. Judging by the data obtained, the entrapment of insulin in RBCs does not offer significant advantages over its free form, but the existing data are insufficient for final conclusions to be drawn.

### 3.4. Erythrocytes Containing Blood Coagulation Factors

In 1979, Goldsmith et al. studied the possibility of loading coagulation factors IX and X into RBCs [141]. These factors were encapsulated into the RBCs of healthy volunteers and patients with deficiencies of IX and X factors by simple reversible hypoosmotic lysis. Despite the fact that factors IX and X are proteins, the obtained CEs were not bioreactors, since they showed procoagulant activity only after the destruction of the cell membrane and release of coagulation factors into the external environment.

In the work of Sinauridze et al., the pharmacokinetics of free factor IX and factor IX incorporated in autologous RBCs was studied in healthy volunteers [142]. The authors suggested that CEs can maintain a significant level of incorporated factor in plasma due to the natural hemolysis of loaded cells in blood vessels at a low rate. It was shown that encapsulation of factor IX into RBCs prolongs its circulation in the bloodstream by 5–10 times compared with a free factor administered intravenously (t_1/2_ were 73.9 ± 16 and 8.9 ± 5.6 h, respectively). Despite the fact that erythrocytes loaded with factor IX were safe and circulated in the bloodstream for a long time, their anticoagulant activity was not investigated in this work. Thus, further study of possible therapeutic efficacy of these CEs is needed.

### 3.5. Morphine Encapsulation into Erythrocytes

To ensure prolonged postoperative analgesia, morphine was incorporated into the autologous RBCs of patients (using the glucose hyperosmotic pulse method). Blood was mixed with a solution of 50% glucose in a ratio of 1:0.5 and incubated for 30 min. Then the RBCs were washed and incubated with a solution of morphine [19,20,143]. In clinical trials in different patients it was found that morphine loaded into RBCs (RBC-M) was able to provide longer analgesic effects than intravenous free-form morphine (M) (24 h for RBC-M vs. 3.2 h for M). However, the observed side effects in patients receiving morphine in these two forms did not differ [144,145,146].

### 3.6. Nanoparticles and Erythrocytes

Inorganic nanoparticles (NPs), along with RBCs, are increasingly used in medicine, in particular, in the field of drug delivery and diagnostics. NPs have a large surface area per unit volume, and are able to bind to a large number of ligands. This increases their affinity for target molecules. In addition, NPs can have unique optical and magnetic properties that enable magnetic targeting and directional fluorescence imaging of cancer cells in the near-infrared. Artificial nanocarriers (NCs) of a new generation have potential advantages unattainable for RBCs, especially with the development of technologies for the synthesis of NCs. Layer-by-Layer (LbL) technology, which allows obtaining NPs with precisely controlled structure and size using various classes of materials, has become an active area of research [147]. The possibility of precise synthesis control allows designing carriers with the specified almost unlimited properties, functions and geometry (from films to fibers and capsules). The methods for preparing LbL-carriers are different and make it possible to encapsulate various types of molecules, such as antibiotics, growth factors and biosensor substances including hydrophobic compounds with the possibility of controlled release in intravascular and extravascular target-organs [148]. One review [148] discussed the possibility of using LbL technology to create synthetic NCs with encapsulated enzymes. However, the key issue here is the opportunity of transferring the synthesis technology of artificial NCs from the scientific laboratories to the production level for clinical application, since some methods of NCs creating are applicable only for small volumes. Scaling of the production process requires large material and time costs [147,148,149]. Further research in vivo is needed to identify the balance of efficiency/risk ratio and to create a regulatory framework for adjusting the production of artificial NCs. In addition, like other synthetic materials, NPs do not have perfect biocompatibility and biodegradability. They are often rapidly destroyed by macrophages of the immune system, and cannot reach other target organs of therapeutic interest. Injection of artificial carriers can activate the complement system, induce the formation of reactive oxygen species, autophagy, inflammation and other toxic side effects. The review by Parhiz et al. [150] discusses in detail the limitations and undesirable side effects of NCs, including biodegradability and biocompatibility ones. In contrast to synthetic capsules, RBCs are well-studied, can be readily obtained and in many ways represent ideal biocompatible and biodegradable drug carriers for intravascular delivery. A combination of these two delivery systems is a promising approach. In this case, the encapsulation of nanoparticles into RBCs creates a “camouflage” for them against the immune system [151]. One review [152] described, in detail, the types of nanoparticles that were already associated with the surface or loaded inside RBCs, as well as the prospects for their use in antitumor therapy.

Muzykantov’s et al. proposed the promising use of synthetic and natural carriers tandem for the treatment of acute critical diseases such as acute respiratory distress syndrome (ARDS), pulmonary embolism (PE) and acute ischemic stroke. They presented the concept of RBC-hitchhiking (RH), in which NCs (adsorbed on the RBC membrane) are transfered from RBCs to the first organ downstream of the intravascular injection [153]. The authors obtained impressive results: they showed that optimized RH formulations can safely and powerfully target NCs to chosen organs via select placement of intravascular catheters in animals. For example, intravenous injection of RH increases liposome uptake in the first downstream organ (lungs) by ~40-fold compared with free NCs. Injection of RH-nanogels intra-carotid artery delivers >10% injected NCs dose to the brain, approximately 10-fold higher than the best affinity component targeting the brain (transferrin), which only delivered 1% of the injected dose.

Various nanoparticles incorporated with drugs such as doxorubicin [154,155], valproate [156], fazudil [157] and pravastatin [158] and encapsulated into RBCs were tested. Entrapment of fluorescent silicone nanoparticles (SiNPs) with doxorubicin into RBCs allowed for a four-fold increase in the half-elimination time of doxorubicin from the mouse bloodstream (up to 7.31 ± 0.96 h) compared to such particles without RBCs [155]. The literature describes promising examples of the use of erythrocytes loaded with nanoparticles with unique optical properties, such as photostability and strong fluorescence, for in vivo imaging and tumor photodestruction, fluorescence imaging for tumor surgery and photoacoustic imaging [157,159,160,161,162].

Thus, the combination of artificial and natural carriers of drugs extends the application boundaries for both of them. Two drug delivery systems with unique advantages/disadvantages supplement each other, which opens up their new multifunctional capabilities. The use of RBCs for the delivery of artificial NCs significantly increases the efficiency and safety of the latter, which can lead to an increase of the benefit/risk ratio and trigger the expansion of NCs production with access to clinical practice. However, there are limitations of this concept because NPs can affect RBCs, as was shown in [163,164]. Adsorption of NPs onto RBCs can lead to an increase in the RBCs stiffening and sensitize RBCs to damage by osmotic, mechanical and oxidative stress. Therefore, it is important to optimize the composition and properties of NCs (NPs) and to perform a detailed analysis of the modified RBCs for their proper use in tandem. To date, RBCs remain the most attractive system for drug delivery due to their easy preparation, complete biocompatibility and biodegradability and the ability to circulate in the bloodstream for a long time.

## 4. Erythrocytes for Targeted Drug Delivery

The targeted delivery of drugs using RBCs can be carried out, firstly, to the cells of the RES (macrophages), as well as in the liver and spleen, i.e., in the body cells, that remove old and damaged RBCs. Thus, this approach may be successfully used to treat tumors of these tissues. To deliver the erythrocyte loaded with the drug into these target cells, it must be modified so that the target cells perceive it as being damaged. There are various methods of such modification. All of them lead to a modification of the erythrocyte membrane. This may be the opsonization of RBCs with antibodies to their membrane determinants (for example, by rhesus-antibodies [165]) or the binding of the complement component C3b to them, since there are receptors for the Fc fragment of IgG and for C3b on the cell surface. Treatment of RBCs with calcium ionophore leads to phosphatidylserine exposure on their surface [166], and treatment with glutaraldehyde cross-links the amino groups on the membrane surface, which makes the cell more rigid. Another method is treatment with reagents that cause clustering of the band 3 protein, for example, by a bifunctional amine–amine cross-linking agent, bisulfosuccinimidyl suberate (BS^3^) in ZnCl_2_ medium [167,168], which leads to the binding of Hb and proteins of the membrane and fixation of complement components on the cell surface [169]. Inactivation of intracellular hexokinase is also described, which leads to disruption of the cell metabolism and a decrease in the concentration of ATP necessary for cell survival [170].

### 4.1. Methotrexate

Methotrexate (MTX) is one of the cytostatic preparations (see above). In 1978, Zimmermann et al. were among the first to demonstrate, in mice, the advantage in the distribution of the erythrocytic form of methotrexate (MTX-RBC) in the body over the free form for intravenous administration. The authors encapsulated the drug by electroporation (i.e., created pores in the RBC membrane using an electrical impulse) through which methotrexate (MTX) penetrated the cell. When this form of the drug was administered to mice over 10 min, almost all the methotrexate that was administered in RBCs (0.75–1.0 doses) accumulated in the liver of animals, while in control experiments (with the introduction of the free form of methotrexate), only 0.25–0.3 of the administered dose accumulated [171].

DeLoach and Barton encapsulated methotrexate in erythrocytes by hypoosmotic methods and showed in dogs, in vivo, that in this case, the drug quickly leaves the RBCs. Thirty minutes after the injection of MTX-RBCs into the bloodstream, 50% of methotrexate appeared free in the plasma [172]. To slow the release of the drug from the cells, treatment of carrier erythrocytes with glutaraldehyde was proposed, which provides an additional advantage, since, as was shown in dogs, 50% of CEs treated with glutaraldehyde are rapidly detected in the liver, i.e., targeted delivery of methotrexate to RES occurs [172,173]. Another method for incorporating methotrexate into RBCs uses the pulse of a hyperosmotic glucose solution. In this case, the cells are incubated for 40 min in a 50% glucose solution. Then, they are gently washed and incubated for 30 min with a solution of methotrexate under normal tonicity. The half-life of such CEs with methotrexate was almost 3.5 times longer than for the free form of the drug (13.5 and 3.9 h, respectively) [174]. In addition, the peak plasma concentration of methotrexate after MTX-RBC administration was lower than with free MTX, but it decreased more slowly. A gradual release of the drug from RBCs was observed. In another study [175], *N*-hydroxysuccinimide biotin ester (NHS-biotin) was bound on the surface of CEs for targeted delivery of MTX-RBCs to the liver. In vivo experiments on rats showed that 1 h after administration of biotinylated MTX-RBCs to animals, 37.2% of biotin appears in the liver, which is almost three times more than after administration of free MTX (11.7%) and almost 1.8 times more than for non-biotinylated cells (20.4%). In an earlier work [176], the same authors modified MTX-RBCs with trypsin (Tt) or phenylhydrazine (PhT) to desialize the cell surface and induce hemichrome in cells, respectively. These two approaches were equally used for the recognition of erythrocytes by macrophages in order to deliver methotrexate for the treatment of RES tumors. Surface-modified erythrocytes loaded with MTX 1 h after administration to animals showed an increased level of methotrexate in the liver compared with the free form of the drug (approximately six times) and with unmodified cells (approximately two times). Phagocytosis by macrophages of surface-treated MTX-loaded erythrocytes was increased by three–five and five–six times for Tt- and PhT-treated CEs, respectively, compared with untreated CEs [176].

The presented examples demonstrate promising possibilities of using erythrocytes for targeted drug delivery to the liver and RES.

### 4.2. Erythrocytes-Carriers for Treatment of Retroviral Infection

Retroviruses are a family of RNA viruses that primarily infect vertebrates. The most famous and actively studied representative of retroviruses is the human immunodeficiency virus (HIV). Currently, nucleoside analogs, which are inhibitors of reverse transcriptase (after anabolic intracellular phosphorylation), are essential components of highly active antiretroviral therapy (HAART). The most famous of these are azidothymidine (and its analogs), dideoxycytidine and other 2′,3′-dideoxynucleosides [169,177]. Furthermore, the antiviral activity of reduced glutathione (GSH) against RNA and DNA viruses is well known. This activity is realized by interfering with protein-envelope folding and by blocking cell transcriptional factor (NF-kB) activation, which decreases the virus transcription and replication [178,179]. The nucleoside analogs protect lymphocytes, but cannot enter macrophages, while GSH inside specially modified RBCs can be captured by macrophages and protect them against viral infection.

Thus, to treat this immunodeficiency, both CEs containing antiretroviral drugs and CEs containing GSH or GSH + antiretroviral drugs can be used, since GSH-loaded RBCs has been shown to provide significant additional effects compared to monotherapy with antiretroviral drugs (nucleoside analogues) [178].

Since the 1990s, Magnani et al. has been actively developing CEs for the treatment of the human immunodeficiency virus. Since the targets and reservoirs of human immunodeficiency infection are cells of the monocyte/macrophage line, attempts have been made to deliver antiretroviral drugs directly to macrophages to prevent transmission of HIV from already infected macrophages to target lymphocytes [180]. The most popular nucleoside analogues, such as 3’-azido-2’,3’-dideoxythymidine and 2’,3’-dideoxycytidine (ddCTP), were encapsulated into RBCs.

It was shown [169] that for the manifestation of pharmacological activity, dideoxynucleosides must be phosphorylated to 5’-triphosphate by cell kinases. Different types of cells within the same species have different abilities to phosphorylate these compounds. To reduce the toxicity of nucleoside analogues, as well as to overcome the problem of the effectiveness of their phosphorylation, Magnani et al. incorporated ddCTP into RBCs in an active phosphorylated form (by the method of hypoosmotic dialysis). For targeted delivery of such RBCs to macrophages, the loaded cells were treated with a bifunctional amine–amine cross-linking agent BS^3^ in ZnCl_2_ medium. This makes the RBCs tougher and induces the binding of autologous immunoglobulin G (IgG) and complement component C3b on the cell surface. Such RBCs are recognized by macrophages and actively phagocitosed. In vitro and in vivo, it was shown that erythrocytes treated in this way loaded with phosphorylated ddCTP were able to significantly reduce typical symptoms of the disease within 3 months [169,181,182,183,184]. The ability to release 3’-azido-2′,3’-deoxythymidine (AZT) from erythrocytes loaded with the azidothymidine derivative di-(thymidine-3’-azido-2’,3’-dideoxy-d-β-riboside)-5’-5’-p1-p2-pyrophosphate (AZTp2AZT) has also been demonstrated in vitro. This prodrug is converted inside erythrocytes into the pharmacologically active AZT by sequential hydrolysis and dephosphorylation [185].

In [179,186,187], interesting results of combination therapy using oral AZT, AZT + DDI (2′,3′-dideoxyinosine) and the additional administration of erythrocytes encapsulated with GSH in each case were demonstrated. The experiments were performed on mice infected by the retrovirus complex (LP-BM5). Studies have shown a decrease in proviral DNA in the brain by about 50% with AZT + DDI treatment and 85% when GSH-loaded RBCs were added to AZT + DDI therapy. For bone marrow, this decrease was about 37% and 60%, respectively [187]. The addition of GSH-loaded RBCs to AZT monotherapy decreased proviral DNA in bone marrow by 60% [186].

RBCs encapsulated with fludarabine have become another possible approach for treating HIV-1 infection. As mentioned above, long-living macrophages in the infected body are the reservoir for the HIV-1 virus. It was shown that chronic infection of human macrophages with this virus increases the expression and phosphorylation of the protein STAT1, which is included in the regulation of many macrophage functions, including cell growth and proliferation [188]. The nucleoside analogue of 9-(β-d-arabinofuranosyl)-2-fluoroadenin-5’-monophosphate (FaraAMP, fludarabine) is active against STAT1-expressing cells and, in culture, is able to kill HIV-infected macrophages, but not uninfected cells. To direct fludarabine to macrophages, it was encapsulated into RBCs, which were then processed by the method described in [169], which causes clustering of the band 3 protein. The final concentration of fludarabine in macrophages after a single 18-h exposure with erythrocytes loaded with fludarabine was estimated at 10–20 μM. In that study, a powerful (>98%) and long-lasting (at least 4 weeks) effect of inhibiting the release of the virus from HIV-infected macrophages was obtained [189].

### 4.3. Drugs Loaded into RBCs for the Treatment of Hepatitis C

To enhance the effectiveness of the therapeutic effect of drugs used in the treatment of hepatitis C, and to minimize their side effects associated with an increase in the dose of drugs, Skorokhod et al. were searching for new ways to simultaneously deliver interferon (INF-α) and ribavirin (RIBA) to the liver [189]. Both drugs were loaded into human RBCs (RBCs-INF-α-RIBA) by the method of hypoosmotic reversible lysis. Cells were opsonized for targeted delivery to macrophages and liver. The entrapment efficiency was 40%. It was shown that RBCs-INF-α-RIBA were stored for up to 3 days at 4 °C without loss of antiviral activity. In vitro, monocyte activation by RBCs-INF-α-RIBA was also demonstrated, as well as the induction of surface receptors of the major histocompatibility complex type II (MHC class II) and Fc receptors that activate cell phagocytic activity. The authors argue that encapsulating INF-α and RIBA into RBCs and targeting the liver helps: (1) to release large amounts of INF-α and achieve higher therapeutically effective concentrations in the liver; (2) to induce autocrine stimulation of macrophages of the liver (and spleen) using INF-α to enhance cellular antiviral protection; (3) to control viral proliferation in macrophages. In this regard, it is advisable to further study a potentially therapeutically effective system in animals.

Forezesh and Zarrin proposed encapsulating a more modern hepatitis C drug, boceprevir, into RBCs in addition to interferon and ribavirin [190].

### 4.4. Macrophage Depletion

It is known that macrophages play an important role in the regulation of numerous biological processes in the body. In addition, it has been repeatedly shown that macrophages contribute to the development of pathologies such as autoimmune hemolytic anemia, immunothrombocytopenia, rheumatoid arthritis and sepsis, and play a key role in the spread of viruses in HIV infections [191]. Tumor-associated macrophages create favorable conditions for cancer progression, promoting angiogenesis and metastasis [192,193,194]. Rossi et al. studied the possibility of temporary depletion of macrophages by incorporating bisphosphonates (clodronate, zoledronate) into RBCs and the targeted delivery of such carriers to macrophages. They showed that RBCs loaded with zoledronate are able to deplete macrophages both in vitro and in vivo [160]. Balb/C mice were injected with 59 mg/mouse of zoledronate encapsulated into RBCs. For targeted delivery to macrophages, loaded erythrocytes were incubated in medium with BS^3^ and ZnCl_2_. After a single injection of encapsuled erythrocytes, macrophage depletion was 29% and 67% for liver and spleen macrophages, respectively.

Another study evaluated the effect of macrophage depletion to prevent Langerhans islet cell allograft rejection in diabetes mice [195]. Graft survival was 19–20 days for control groups of mice receiving unloaded erythrocytes or saline, 25 days for mice receiving free clodronate and 35 days for mice receiving clodronate in RBCs.

### 4.5. Antigens Loaded into Erythrocytes or Associated with Their Surface

#### 4.5.1. Immunization

Binding antigens to the surface or encapsulating them inside the carrier erythrocytes opens up new possibilities for using such erythrocytes for immunization as an alternative to adjuvants (substances that adsorb antigen on their surface), namely, the possibility of delivering antigens directly to the immune system into antigen-presenting cells—macrophages or dendritic cells (DCs). Dendritic cells are believed to be most effective in initiating antigen-specific responses, but macrophages are also able to facilitate the presentation of peptides to T lymphocytes [196]. Magnani et al. has repeatedly shown that protein antigens (bovine serum albumin, porcine liver uricase, yeast hexokinase) and glycoproteins B of herpes simplex virus type 1 (HSV-I), which are associated with the surface of autologous RBCs via the biotin–avidin–biotin bridges, induce a higher immunological response (higher antibody levels) in mice than the response obtained using Freund’s adjuvant, which is often used in immunization [197,198]. Later, it was shown that the HIV-1 Tat protein, linked through the biotin–avidin–biotin bridges to the erythrocyte surface (RBC-Tat), has immunotherapeutic potential. This protein is important for virus replication and infectious activity (the presence of antibodies against Tat correlate with slower progression of the disease). Tat protein is immunogenic [199]. Erythrocytes associated with Tat (RBCs-Tat), in amounts 250 times less than the amount of soluble Tat in Freund’s adjuvant, are capable of eliciting specific responses of anti-Tat T killers. Moreover, the production of Tat neutralizing antibodies was observed in six out of six mice, in contrast to two out of six mice for Tat in Freund’s adjuvant.

In other works [200,201], using bacterial toxoids, proteins and enzymes as antigens, it was shown that immunization is also possible by encapsulating antigen in RBCs. In B6D2F1 and Balb/C mice, the total titers of specific antibodies (binding, lysing and neutralizing the antigens) and only neutralizing antibodies against introduced antigens were several times higher during immunization with antigens loaded into RBCs than after immunization with free forms of antigens [200].

#### 4.5.2. Cancer Immunotherapy

Cancer immunotherapy is the use of the immune system to kill tumor cells that have specific tumor-associated antigens (TAA) [202]. Banz et al. proposed a strategy for using RBCs loaded with tumor-associated antigens in cancer immunotherapy. Immunization against TAA induces TAA-specific cytotoxic T lymphocytes (CTLs), which are capable of controlling tumor growth. Efficient and targeted delivery of TAA in vivo to DCs can be effective in tumor immunotherapy since it induces strong CTLs responses against the tumor [203]. It was shown in mice [204] that erythrocytes bearing an antigen (in this case, ovalbumin) in combination with polyinosine–polycytidylic acid (Poly (I:C)) introduced intravenously, can be effectively captured by antigen-presenting cells (APC). This causes antigen-specific responses of CD4^+^ and CD8^+^ T cells, which are able to induce in vivo ovalbumin-specific cell lysis even 30 days after CEs administration. Ovalbumin was loaded into RBCs by hypoosmotic dialysis (RBC-OVA). To enhance the phagocytosis of these erythrocytes with antigen-presenting cells, they were treated externally with antibodies (anti-TER119 mAb), and then were administered to C57BL/6 mice intravenously. RBC-OVA was mixed with Poly (I:C) before injection to enhance the induction of T-cell responses, as Poly (I:C) is a toll-like receptor III ligand that activates the CD4^+^ T cell response specific for alloantigen of RBCs [205,206]. Ninety minutes after injection of RBC-OVA + Poly (I:C) to mice, phagocytosis of the introduced RBCs by antigen-presenting macrophages and dendritic cells was observed.

The effectiveness of such a tumor-associated antigen delivery system was also demonstrated in two models of mice with melanoma [207]. The artificial ovalbumin antigen or tyrosinase 2 protein antigen (TRP-2) was encapsuled into red blood cells and tested on E.G7-OVA and B16F10 tumor models, respectively. The administration of a small amount of tumor-associated antigen (TRP-2) loaded into RBCs treated with antigen anti-TER119 in combination with Poly (I:C) caused an antigen-specific T-cell response and tumor growth control in mice, whereas the same amount of free TRP-2 did not cause a similar response.

#### 4.5.3. Induction of Immune Tolerance

The opposite of immunization is the stimulation of immune tolerance, that is, the “training” of the immune system to create tolerance (resistance) to a particular antigen in order to prevent its attack. Such stimulation can be used in autoimmune diseases, when the immune system attacks its own antigens, during an allograft transplant, or in case of an allergy to a drug used in therapy. The induction of immune tolerance is often carried out using molecules that inhibit the immune system, such as rituximab (anti-CD20 monoclonal antibody), cyclophosphamide, and methotrexate, or by depleting B cells necessary for the immune response. In [208], the authors proposed the use of erythrocytes for the induction of immune tolerance. The drug, to which it was necessary to induce immune tolerance, was loaded into RBCs. That study showed that the drug inside the cell-carriers does not interact directly with antibodies, which may be present in plasma. The authors investigated the possibility of obtaining immune tolerance in mice for the enzyme alglucosidase α (AGA), a recombinant analogue of acidic α-glucosidase, which is currently used in enzyme replacement therapy for Pompe disease (glycogen storage disease caused by α-glucosidase deficiency). For targeted delivery of the drug to antigen-presenting cells of the liver and spleen, the erythrocytes loaded with the enzyme were treated with BS^3^/ZnCl_2_.

As mentioned above, therapy for Pompe disease is carried out by frequent intravenous administration of AGA, which ultimately causes a stable humoral response and leads to the need to discontinue treatment. This work showed that erythrocytes encapsulated with AGA and then BS^3^/ZnCl_2_-treated have tolerogenic properties, i.e., they are able to eliminate the humoral response to AGA and restore tolerance to replacement therapy. First, the mice were injected intravenously with AGA-loaded RBCs (three times) and then they were sensitized to AGA using different adjuvant molecules. Control animals received free AGA instead of the encapsulated molecules. A strong decrease in the specific humoral response was observed in the experimental group one-week after treatment with AGA-loaded RBCs. This effect was maintained for at least two months without affecting the overall immune response [208].

The effectiveness of the induction of immune tolerance depends on several factors, such as the route of administration and the dose of antigen (Ag), as well as the type of target antigen-presenting cells [209]. DCs and macrophages ingest foreign antigens and present fragments of these antigens on their own surface for recognition by T cells, and thereby, participate both in the induction of immunity and in the stimulation of its tolerance. After B or T cells recognize Ag on the APC surface, the choice between tolerance and immunity depends on the amount and type of Ag, type of APC and the number of co-stimulation molecules CD80 and CD86 (which bind to the CD28 receptor on the membrane of T-lymphocytes) on the DCs’ surface. The maturation status of DCs is a key factor in the development of immunity or the induction of tolerance. Mature DCs induce immunity, while immature DCs induce tolerance, since they are capable of expressing low levels of MHC class II surface antigens and costimulatory molecules, which are necessary for the antigen presentation to T-lymphocytes [210]. The presentation of antigen to T-lymphocytes, in turn, stimulates the differentiation of immature T-lymphocytes into cytotoxic CD8^+^ cells or CD4^+^ helper cells. The liver plays an important role in the induction of tolerance due to its specific composition of antigen-presenting cells. Liver DCs have an immature phenotype and, therefore, are not able to elicit an Ag-specific T-cell response, but induce the development of T-cell tolerance. Several subpopulations of DCs of the spleen are also involved in the induction of tolerance [211]. Thus, the delivery of Ag to the corresponding DCs of the liver and spleen is an attractive strategy for the induction of specific antigen tolerance.

An example is the work of Cremel et al., which demonstrated the possibility of inducing immune tolerance in mice by administration of RBCs loaded with ovalbumin (OVA) as antigen and treated with calcium ionophore or BS^3^ [209]. It was shown that intravenous injection of such erythrocytes into mice sensitized to ovalbumin caused a strong decrease in specific humoral and cellular immune responses (the appearance of 19%–22% of activated OVA-specific CD8^+^ T cells vs. 58%–64% for mice without the induction of immune tolerance). Such a response was observed during, at least, 34 days after the induction of tolerance and was antigen-specific, without causing complete suppression of the immune system.

ERYTECH Pharma has patented both methods of using erythrocytes as carriers of antigens—in cancer immunotherapy to stimulate a cytotoxic cell response directed against tumor cells expressing an antigen [212], and as a system that induces a specific immune tolerance to enzymes, which are used in enzyme-replacement therapy of diseases such as, for example, Pompe disease, Fabry disease, mucopolysaccharidosis, hemophilia A and B, rheumatoid arthritis, multiple sclerosis, etc., requiring stimulation of the immune tolerance to achieve a therapeutic effect [213].

## 5. Carrier Erythrocytes in the Diagnostics

### 5.1. Contrast Agents in Magnetic Resonance Imaging

MRI is a non-invasive method for visualizing the structure and function of tissues, which is widely used in clinical practice. Despite the fact that MRI allows high-resolution anatomical images to be obtained, the possibilities of this method can be significantly expanded with the help of contrast agents. They are used to improve the differentiation of malignant and healthy tissues [214], as well as for MR angiography, which reveals damage to blood vessels, primarily myocardial damage, atherosclerosis, thrombosis, aneurysms and other vascular diseases [215]. The localized interaction of contrasting agents with protons of water molecules in various tissues creates a contrast by decreasing the time of their longitudinal (T_1_) and transverse (T_2_) relaxation (the time during which the protons return to their equilibrium state after exposure to an electromagnetic pulse). This relaxation is different in healthy and pathological tissues, and depends on the surrounding molecules and atoms. Based on this difference, MRI images are constructed. Paramagnetic and superparamagnetic contrasting agents increase relaxation rates (1/T), thereby enhancing contrast. A measure of the sensitivity of the contrast agent is its longitudinal and transverse relaxivity (r_1_ and r_2_, respectively), which show how the corresponding relaxation rate changes as the concentration of the contrast agent changes (C) (see Equations (1) and (2)):1/T = r × C(1)
r = 1/(T × C)(2)

Various metal derivatives, primarily gadolinium oxides, chelate complexes of lanthanides and gold nanoparticles, as well as superparamagnetic iron oxide nanoparticles (SPIO) and ultrafine superparamagnetic iron oxide nanoparticles (USPIO), can be used as contrasting agents. However, the use of these nanoparticles in MR angiography is limited, since the surface of nanoparticles undergoes opsonization upon intravenous administration, i.e., it adsorbs plasma proteins. This stimulates and facilitates phagocytosis of these particles, so that their half-life in blood is 1–3 h (a decrease in the size of nanoparticles increases the half-life), and the time interval for observation after bolus administration of the drug is only a few minutes [216,217,218,219,220]. To date, as a result of these reasons, many SPIO and USPIO preparations in Europe and the USA are practically not used [221,222]. On the other hand, due to the selective uptake and accumulation in RES cells, superparamagnetic iron oxide nanoparticles have become very popular for imaging the liver and spleen [223,224].

In 2008, Antonelli et al. proposed the encapsulation of magnetic nanoparticles based on iron oxide in RBCs in order to increase their lifetime in circulation [225]. This group has a large number of works devoted to the study of the properties of various magnetic nanoparticles based on iron oxide, both newly created and commercially known, such as Resovist (Bayer Schering Pharma), Sinerem and Endorem (Guerbet, France) etc., loaded into RBCs [222,225,226,227,228]. It has been shown that not all iron oxide-based nanoparticles can be successfully incorporated into erythrocytes. The result depends on properties of nanoparticles, such as their size, nature of the dispersant and surface charge, that are important to obtain monodispersed nanoparticles in suspension, as well as on the chemical properties of particle surface coating [197]. On the other hand, SPIO encapsulation in RBCs can increase the circulation time of these particles in the bloodstream by up to 12 days, which makes it possible to use them in MR angiography for long-term imaging and long-term monitoring of cardiovascular diseases [229].

It was demonstrated in [230] that the encapsulation of USPIO nanoparticles into RBCs leads to an increase in their transverse relaxivity r_2_ and a very high ratio of relaxivities r_2_/r_1_, which makes them promising for use as a negative contrast agent in the blood pool. Other studies have also demonstrated the advantages of using carrier erythrocytes rather than suspensions for gadolinium oxide nanoparticles [231], chelate complexes of lanthanides [232] and gold nanoparticles [233] as contrasting agents for MRI.

### 5.2. Blood Analyte Biosensors

For long-term non-invasive in vivo monitoring of certain blood parameters (analytes), such as glucose concentration or pH, Ritter et al. proposed the use of RBCs loaded with a fluorescent dye that responds to changes in the concentration of an analyte in the bloodstream [33,234,235,236]. In this case, autologous RBCs encapsulated with fluorescent dyes (sensors) are introduced into the patient’s circulation for analytical monitoring. The fluorescent signal of the erythrosensors can be excited and detected non-invasively through the skin when excited by an external light source in the visible wavelength range (for example, a laser diode). It was shown in [34] that erythrocytes loaded with fluorescein isothiocyanate (FITC), a pH-sensitive fluorescent dye, have an excellent ability to reversibly monitor in vitro pH in the physiological range with a resolution of up to 0.014 pH units. According to the authors, the fluorescence intensity increases with increasing extracellular pH, since RBCs quickly balance pH with the external environment through a chloride–bicarbonate exchanger. However, it turned out that for pH measurements in vivo in the physiological range, the sensitivity of such a system is too low. Thus, the next step to facilitate the use of RBCs as biosensor carriers should be the development a fluorescent sensor with higher sensitivity and optimization of RBCs loading to obtain a higher signal level [35].

## 6. A Novel Trend in the Use of Red Blood Cells as a Delivery System

To use erythrocytes to deliver drug compounds, these compounds must be loaded into cells. There are many different loading procedures, which have been developed for a long time and continue to improve. Most often for this the RBC membrane is subjected to certain physical influences. Despite the fact that the process is carried out under conditions which spare the cell, such procedures, of course, reduce the quality of the resulting loaded cells compared to the original erythrocytes [237]. In addition, the effectiveness of the encapsulation of a protein depends on its size and other physical properties, and is far from being always sufficient. Against this background, the newest trend of using RBCs as carriers of certain enzymes looks very interesting.

RubiusTherapeutics (Boston, USA) combined the successes of genetic engineering and the unique properties of RBCs by developing a new class of cell drugs, which they called Red Cell Therapeutics™ (RCTs) [238]. RCTs are allogeneic erythrocytes that express targeted biotherapeutic proteins (enzymes) inside or on the surface of the cell. To obtain such RCTs, allogeneic hematopoetic progenitor cells (CD34^+^) are first genetically modified using a gene cassette or lentiviral vector to provide expression of one or more targeted therapeutic proteins. The converted cells are placed in a bioreactor for their further maturation up to reticulocytes. The resulting cells have the same characteristics as normal RBCs and contain, inside or on their surface, the target therapeutic protein for the treatment of the suspected disease. Such RCTs can be used in enzyme-replacement and anticancer therapy (cancer immunotherapy), as well as in the treatment of autoimmune diseases. Currently, the first phase of clinical trials of RTX-134, erythrocytes carrying the *AvPAL* gene inside cells (the phenylalanine-ammonia lyase gene *Anabaena variabilis*), is being conducted to treat adult phenylketonuria (NCT04110496) [239]. At conferences in Philadelphia and Boston in 2019, Zhang [240] and Moore [241] proposed interesting ideas for creating artificial antigen-presenting cells, the genetically modified erythrocytes (RCT-aAPC), which expresses immunomodulating signals that are directed against the tumor. Such cells, on the one hand, are loaded with tumor-specific antigen and costimulatory molecules, and, on the other hand, express proteins of the main histocompatibility class I complex on the surface to create an effective tumor-specific T-cell response. Using this strategy in mice showed 60% inhibition of tumor growth on day 7 after administration of RCT-aAPC to animals.

Thus, RubiusTherapeutics technology represents a new promising approach for the delivery of therapeutic substances to patients using erythrocytes. These results are especially encouraging in light of the fact that, in 2017, a method was developed to create an “immortal” line of erythrocytes from the corresponding erythrocyte precursors [242].

If you have a culture of unipotent erythrocyte precursors, you do not need to worry about managing their differentiation. However, unlike stem cells, the number of divisions of such cells is limited; thus, they must be immortalized, i.e., modified so that their division can be endless. For this, bone marrow, cells were genetically modified by adding a human papilloma virus gene to them, which allows cells to divide unlimitedly. Then, the transition of the modified cells into erythrocyte precursor cells was induced. Thus, a new cell line, BEL-A (Bristol Erythroid Line Adult), was created. The course of these cells’ differentiation did not differ from the corresponding stages of development of pluripotent stem cells. The results obtained appear promising for the possibility of scaling the process to obtain the desired RBCs in sufficient quantities.

## 7. Limitations of the RBCs’ Use as Drug Carriers

Despite the fact that RBCs are very promising for use as drug carriers, their use has a number of limitations. The source of RBCs is blood; thus, the use of allogeneic blood can lead to errors in choosing the right blood type and to the transmission of various infections. However, these disadvantages are common to all transfusion of blood products. These situations are very rare, and currently they are not the principal barrier to transfusion of any blood products, including erythrocytes loaded with drugs. In addition, production of carrier erythrocytes are associated with the need for sterile work and the complexity of the large-scale production of such cells. Creating automatic devices can solves these problems. Another disadvantage is related to the fact that if any crude method was used for CEs preparation, the quality of the resulting cells may not be high enough. In this case, these CEs will rapidly degrade in the bloodstream, and the drug may be released uncontrollably. This complicates drug delivery and can lead to adverse side effects. However, the methods currently used are soft enough and do not have a strong effect on RBCs.

There are also other restrictions. The first of them is that far from any substance can be incorporated into RBCs. Some low molecular weight compounds that easily pass through the erythrocyte membrane are not only easy to enter, but also just as easy to leave the cells, which makes it impossible to create a long-term depot form of these compounds based on RBCs in the bloodstream [82,94,140]. To slow the release of such substances from RBCs, the cells may be treated with different crosslinking agents (primarily for NH_2_– or HS– groups on the membrane surface). This may be glutaraldehyde, BS^3^, etc. [166,167,168,169]. However, although this slows the release of drug compounds from the cells, the membrane of such erythrocytes changes so much that they are quickly recognized by RES cells and removed from the bloodstream. Another way to retain a therapeutically effective substance that easily passes through the erythrocyte membrane inside the cell is to encapsulate a prodrug in the erythrocytes, for example, a phosphorylated form of this compound, which cannot pass through the cell membrane but can be dephosphorylated by phosphatases of RBCs, turning it into a therapeutically active substance that gradually leaves the cells. The opposite situation is also possible when for activation, the substance must be phosphorylated inside the erythrocyte by the corresponding erythrocyte phosphokinases (as in the case of dideoxynucleotides [169]). In all these cases, the limitation of the use of RBCs as drug carriers is that the activity of the desired enzymes in the cells of different patients can vary greatly, which does not allow to obtain stable results [243].

If RBCs are supposed to be used as bioreactors, then in a number of cases a second serious limitation arises. This is due to the possible effect of the loaded enzymes on the erythrocyte metabolism, primarily glycolysis. This overlap can lead to depletion of the pools of some metabolites (for example, NAD(P) and NAD(P)H) if they are used simultaneously by glycolysis and enzymes built into RBCs. In this case, a stationary state can be lost in glycolysis, which leads to rapid cell death in the bloodstream (Protasov et al., unpublished data). A possible way to deal with this situation may be to calculate the permissible doses of the loaded enzymes, which do not yet lead to the loss of a stationary state in glycolysis (using mathematical models). Moreover, it is possible to encapsulate the necessary cofactor and the target enzyme into RBCs together (provided that cofactor cannot quickly leave the cell). Sometimes, the work of the enzyme inside the RBCs may be limited by the rate of transport for the necessary substrate of the reaction into the cells. This happened, for example, when ammocytes based on glutamate dehydrogenase [59,60,72] or glutamine synthetase [73,74] were created. In this case, the researchers proposed for incorporation into the RBCs a new enzyme system consisting of two enzymes that provided cyclic consumption and production of the necessary metabolites inside the cell. This made the process independent of the transport of these metabolites [75]. Another area of modern developments to improve the delivery of drugs that can affect the metabolism of RBCs is associated with the replacement of RBCs with artificial RBCs or hybrid nanoparticles, the surface of which contains fragments of the RBC membrane, to ensure their long lifetime in the bloodstream [9]. However, these are only scientific developments, which are far from clinical use.

Thus, there are real restrictions on the use of RBCs as drug carriers; however, they can be circumvented in many cases, both by improving experimental methods of work, and by using mathematical models of CEs to properly account for the effects of the loaded compounds on RBCs metabolism.

## 8. Conclusions

Drug delivery using natural biological carriers is a fast-developing field. Due to their unique biophysical properties, erythrocytes have great potential in this area. Recently, their use has been increasingly expanding both in therapy and in the diagnosis of many diseases. The use of carrier RBCs is very important to prevent unwanted immune responses after the introduction of protein molecules, especially if repeated administration of these drugs is required. RBCs are able to provide the necessary protection for the protein preparation from the immune system and plasma proteases, increasing the lifetime of the drug in the bloodstream, and thereby enhancing its therapeutic effect. In addition, special processing of the membrane of encapsulated RBCs allows targeted delivery of drug-loaded cells to macrophages, dendritic cells, liver and spleen, which is also increasingly used in various fields of medicine. In the case of a number of cytotoxic drugs, the greatest gain when loading the drug into RBCs is achieved due to the fact that, as has been proven, RBCs allow the prolongation of a drug’s therapeutical effect due to its gradual release into the bloodstream. Simultaneously, reducing the peak concentration of free drug in plasma is achieved during administration, which is associated with a decrease in negative side effects, such as cardiotoxicity, with the introduction of anthracycline antibiotics. In Table 1, we collected the drugs and substances encapsulated into RBCs since 1973.

Despite such positive properties and the widespread popularity of carrier erythrocytes in scientific research, only a few drugs loaded into RBCs have now reached clinical use. Perhaps this is due to the complexity of scaling the production of such drugs, since therapy with RBCs incorporated with drugs is more likely to be personalized medicine and requires an individual approach. However, there are two companies that have surpassed all the barriers and are actively promoting this method of drug delivery in clinical practice. These are ERYTECH Pharma (France) and EryDel (Italy). ERYTECH is conducting final clinical trials of erythrocytes loaded with asparaginase (Eryaspase) for the treatment of pancreatic cancer and triple-negative breast cancer [244]. Methionine-γ-lyase loaded into RBCs (erymethionase) for the treatment of solid tumors and the encapsulation of enzymes in RBCs for replacement enzyme therapy and of antibodies for cancer immunotherapy are under development and in preclinical trials.

EryDel, in turn, focused on clinical trials of dexamethasone (EryDex) for the treatment of ataxia telangiestasia [245]. A device developed by EdyDel was also used to prepare thymidine phosphorylase in erythrocytes (EE-TP) for the treatment of mitochondrial neurogastrointestinal encephalomyopathy [246].

Erythrocytes encapsulated with phenylalanine-ammonia lyase for the treatment of phenylketonuria, recombinant uricase for the utilization of uric acid and guanidine methyltransferase for enzyme replacement therapy are currently at the preclinical stage. The European Medical Agency has already granted the status of orphan drugs to dexamethasone phosphate for the treatment of cystic fibrosis [247] and to l-asparaginase for the treatment of pancreatic cancer [248] and acute lymphoblastic leukemia [249].

Thus, it can be expected that in the near future, the carrier erythrocytes of drugs will be widely used, particularly in enzyme replacement and antitumor therapy.

## Figures and Tables

**Figure 1 pharmaceutics-12-00276-f001:**
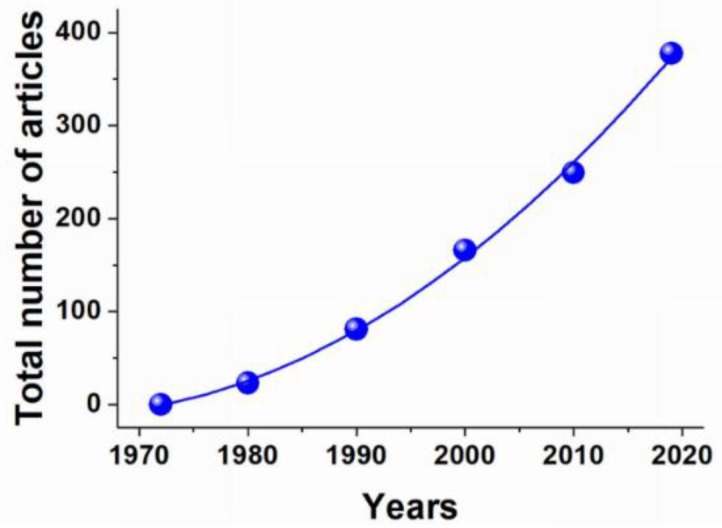
Change in the total number of articles published in the world on the subject of erythrocyte carriers of biologically active substances, since 1973.

**Figure 2 pharmaceutics-12-00276-f002:**
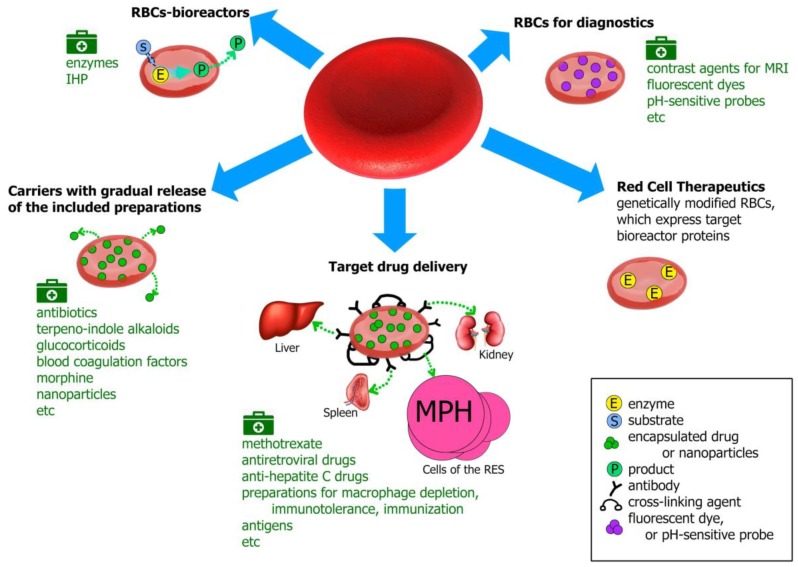
Possible modes of use of RBCs loaded with biologically active compounds and nanoparticles. MRI—magnetic resonance imaging; IHP—inositol hexaphosphate; MPH—macrophages; RES—reticuloendothelial system.

**Table 1 pharmaceutics-12-00276-t001:** Substances that were loaded into erythrocyte.

Active Substance	Application	References
β-Galactosidase	-	[21]
β-Glucocerebrosidase (β-glucosidase)	Gaucher disease	[21,37,38,44,45,46,250]
β-Glucuronidase	Syndrome Slaya	[251]
l-Phenylalanine ammonia lyase	Phenylketonuria	[48,252,253]
Phenylalanine hydroxylase		[50,254]
Uricase (uratoxidase)	Uric acid removal	[255,256]
Urease, urease + alanine dehydrogenase	Urea utilization	[257,258,259]
Adenosine deaminase	Severe combined immunodeficiency caused by deaminase deficiency	[27,55,56,57,58,260]
Thymidine phosphorylase	Mitochondrial neurogastrointestinal encephalomyopathy (MNGIE)	[56,246,261,262]
Glutamate dehydrogenase	Hyperammonemia	[59,60,72]
Glutamine synthetase		[73,74]
Glutamate dehydrogenase + alanine aminotransferase		[11,63]
Arginase	Hyperammonemia due to arginase deficiency	[263]
Alcohol dehydrogenase	Alcohol and methanol intoxication	[59,60]
Alcohol oxidase		[61]
Acetaldehyde dehydrogenase		[62]
Alcohol dehydrogenase + acetaldehyde dehydrogenase		[11,63]
Formate dehydrogenase	Methanol intoxication	[264]
Cyanide sulfurtransferase (rhodanase)	Cyanide intoxication	[65,66,67,68,69,70,265,266]
Catalase, PEG-catalase	Antioxidant	[267]
l-Asparaginase	Antitumor therapy	[16,22,24,83,84,85,86,237,268,269,270,271,272,273,274,275,276,277,278,279,280,281,282,283,284]
l-Methioninase		[87,88,285,286]
Arginine deiminase		[89,90]
Hexokinase, glucose oxidase	To decrease blood glucose (diabetes)	[287,288]
Insulin		[138,139,140,289,290]
Inositol hexaphosphate (IHP)	Sickle cell anemia	[17,18,91,92,93,94,95,291,292,293,294,295,296,297,298,299,300,301]
Methotrexate	Cytotoxic drugs (antitumor antibiotics)	[138,171,172,173,174,175,176,302,303,304]
Mitaxantrone		[112,113]
Doxorubicin, daunomycin		[103,104,105,106,107,108,109,110,111,305,306,307,308,309,310,311,312,313,314,315,316,317,318]
Amikacin	Broad-spectrum antibiotics	[319,320,321,322]
Gentamicin		[323]
Tetracycline		[324]
Penicillin G		[10]
Actinomycin D	Cytotoxic drugs (antitumor antibiotics)	[10]
Cytosine β-d-arabinoside		[10,325]
Carboplatin		[326]
Fluorouracil (5-fluoro-2-deoxyuridine)		[327,328]
Bleomycin		[329]
Vincristine, vinblastine		[120,121,330]
Paclitaxel		[331]
Fludarabine phosphate (2-Fluoro-ara-AMP)	Cytostatic drug (antitumor therapy, HIV)	[118,332,333,334]
Dexamethasone	Glucocorticosteroids (anti-inflammatory drugs)	[10]
Dexamethasone-21-phosphate		[129,130,131,132,133,135]
		[335,336,337,338,339,340,341,342,343]
Betamethasone phosphate		[344]
Prednisolone-21-phosphate		[25,128,345]
Diclofenac	Nonsteroidal anti-inflammatory drug	[346]
Nucleoside reverse transcriptase inhibitors (2,3-dideoxytidine-5-triphosphate (ddCTP), zidovudine (AZT), (AZTp2AZT), didanosine (DDI)) in combination with reduced glutathione (GSH)	Therapy of HIV, retroviral infections	[169,181,182,183,184,185,187,347,348,349,350,351,352]
Fludarabine + AZT + GSH		[353]
Nucleoside protease inhibitors (PNA_PR2_)		[354]
Interferon + ribavirin	Hepatitis C therapy	[189,190]
Ribavirin		[355]
Antigens	Immunization	[197,198,199,200,201,356,357]
	Cancer immunotherapy	[204,207,358]
	Induction of immune tolerance	[208,209,213]
Enalaprilat	Angiotensin-converting enzyme (ACE) inhibitor (arterial hypertension)	[359,360,361]
Morphine	Opioid analgesia	[19,20,143,144,145,146]
Tramadol		[362]
Factors IX and X	Hemophilia	[141,142]
Interleukins 2 and 3	Immunomodulators, antitumor therapy	[363,364,365,366,367,368]
Superoxide dismutase	Antioxidant	[368,369,370,371]
DNA	Gene therapy (gene delivery)	[372,373,374]
Clodronate	Macrophage depletion	[195,375,376]
Zoledronate		[191]
Valproate	Epilepsy	[156]
Phenytoin		[377]
Primaquine	Malaria	[378,379]
Pravastatin	Cardiovascular disease prevention, treatment of abnormal lipids	[158,380,381]
Cyclosporin A, tacrolimus	Immunosuppressants	[36]
Aminazine (Chlorpromazine)	Antipsychotic (in psychiatric practice)	[382]
Naloxone	Opioid receptor antagonist (opioid overdose)	[383]
Ambroxol	Respiratory diseases (fibrosis)	[384]
Superparamagnetic nanoparticles	Contrast agents in MRI	[222,226,227,228,229,230,231,232,233,385,386,387,388,389,390,391]
Nanoparticles	From drug delivery to fluorescence or photoacoustic imaging	[151,152,154,155,156,158,159,160,161,162,392,393,394,395,396,397,398]
Fluorescent dyes (FITC)	Blood analyte biosensors (glucose, pH)	[33,34,35,234,235,236]

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
