# Peer review of "Erythrocytes as Carriers: From Drug Delivery to Biosensors"

_pharmaceutics, 2020, doi:10.3390/pharmaceutics12030276_

Round 1

Reviewer 1 Report

This is a well written and comprehensive review of erythrocyte carriers. I have a few minor comments for the authors to consider:

  1. Line 118. Reference 28 is not the original source of this therapeutic approach and this should be changed to Moran et al Neurology. 71, 686-688 (2008)
  2. Line 38. Please amend this sentence to make it clear that the erythrocytes are manipulated to reversibly form pores and that this is not a physiological characteristic.
  3. Line 99. Only glucocerebrosidase was encapsulated into erythrocytes- please amend the sentence to clarify this.
  4. Line 195. Please make it clear that TRYbeCA1 is the trial name and not a type of cancer. Also, for consistency include the trial name for the breast cancer trial (TRYbeCA2)
  5. Line 270. Could the authors confirm whether this was a regulatory clinical trial or a compassionate use pilot study.
  6. The authors refer to the various encapsulation processes throughout the manuscript. It may be helpful to the novice reader to have a small section at the beginning to give a few sentences on the methods used for encapsulating molecules.
  7. Line 354. This clinical trial is now complete, so please remove the trial number.  Please change the reference from 107 to:  Inflamm Bowel Dis. 2013 Aug;19(9):1872-9. doi: 10.1097/MIB.0b013e3182874065.
  8. Line 510. The clustering has already been described and does not need repeating.
  9. Line 442. Please be consistent with the use of haemoglobin or the abbreviation Hb.
  10. Page 20. Line 14. Please place reference 230 alongside reference 213 as Erydel are not developing EE-TP, but supplying the automated device.
  11. Could the authors ensure that they have correctly cited information and used the correct citations.

Reviewer 2 Report

The authors have described Erythrocytes as carriers: from drug delivery to
 biosensors

  1. The article is nicely organized and written and cover most of the essential parts
  2. I do not see any discussion from the authors about the research that has been undertaken by labs of Vladimir R. Muzykantov and Samir Mitragotri. I would highly recommend the authors to incorporate the literature from the two senior pioneers in the field of erythrocytes as carriers for drug delivery. There are numerous references by both the authors and I would like to see them incorporated before acceptance. 
  3. Also the authors need to discuss about the limitations of Erythrocytes and the approaches that have been tried to date.
  4. Provide expert opinion about the work and the future perspectives

Only once the above changes especially the comment 2 are incorporated I would recommend the acceptance of the publication.

Reviewer 3 Report

The manuscript is a review of applications of erythrocytes as drug and contrast agent carriers as well as microbioreactors able to synthesize specific compounds in the bloodstream. The review is very well written and comprehensive even for a reader that is a bit out of the topic. The strong point is the evaluation of the real applicability of such methods through analysis of literature dealing with in vivo tests and clinical trials. The review is much more detailed than other available in the popular scientific databases, which although quite few of them were published during the last two years, makes it worth acceptance.

I would ask to add few sentences in the introduction on the source of erythrocytes. A short subchapter on the drawbacks of such systems should also be added, for example is there any chance for an uncontrolled release of drugs, why such systems are not yet commonly used? Also it would be interesting to know what is the therapeutic effect of drugs after such encapsulation, the release is prolonged but is the concentration high enough to provide therapeutic effects (e.g. page 6, lines 270-282).

I really appreciate that in many places there are comments why a certain system did not work and what could remedy the situation(e.g. page 4, lines 165-170). That is a very strong point of the work.

Specific comments:

  1. 1 would be more informative if the number of articles would be the total number per year and not a cumulative value.
  2. Page 15, line 720, there is a typo in the name of the French company, a Russian “and”

Reviewer 4 Report

This manuscript reviews erythrocytes as carriers in the field of drug delivery and biosensors. The topic should be able to interest a range of readers. The writing and structure are generally good but still needs improvement. Here are some comments which may help improve the manuscript.

  1. Section “1. Erythrocytes and drug delivery”, according to the content, it is suggested to revisited to be “1. Erythrocytes as carriers”

  1. Figure 1, the resource of the statistics should be described.

  1. As mentioned by the authors, “Since there are separate articles in this Issue devoted to a detailed description of the enzymes loaded in RBCs, in our review, this subject is considered very briefly”. The section "2. Erythrocytes-bioreactors" actually looks long compared with the whole manuscript, which is suggested to be refined.

  1. Please make a schematic summary figure to illustrate the erythrocyte carriers in the field of applications in drug delivery and biosensors, which makes the reviewer clear and attractive to readers.

  1. The English writing needs improvement. Please revise accordingly.

6. Please discuss and compare the erythrocytes to other carriers (i.e., some nanoparticles) in drug delivery.
